# Long Context Transfer from Language to Vision

## Abstract

Video sequences offer valuable temporal information, but existing large multimodal models (LMMs) fall short in understanding extremely long videos. Many works address this by reducing the number of visual tokens using visual resamplers. Alternatively, in this paper, we approach this problem from the perspective of the language model. By simply extrapolating the context length of the language backbone, we enable LMMs to comprehend orders of magnitude more visual tokens without any video training. We call this phenomenon *long context transfer* and carefully ablate its properties. To effectively measure LMMs' ability to generalize to long contexts in the vision modality, we develop V-NIAH (Visual Needle-In-A-Haystack), a purely synthetic long vision benchmark inspired by the language model' s NIAH test. Our proposed Long Video Assistant (LongVA) can process 2000 frames or over 200K visual tokens without additional complexities. With its extended context length, LongVA achieves state-of-the-art performance on Video-MME and MLVU among 7B-scale models by densely sampling more input frames.

## 1 Introduction

Driven by the progress of Large Language Models (LLMs) (Brown et al., 2020; Team, 2023; Touvron et al., 2023; Team, 2024a; Ormazabal et al., 2024; Mistral, 2024; Cohere, 2024), multiple studies are conducted to extend their capability to understand images and videos (Li et al., 2023b; Dai et al., 2023; Team, 2024b; Liu et al., 2023c). With modality alignment and visual instruction tuning, these Large Multimodal Models (LMMs) have shown impressive abilities such as captioning and visual question-answering. While current LMMs have demonstrated promising performance on tasks involving single images and short videos (Song et al., 2024b; Lin et al., 2023a; Maaz et al., 2023; Zhang et al., 2023), effectively processing and understanding extremely long videos remains a significant challenge (Wang et al., 2024b).

One primary reason for this challenge is the excessive number of visual tokens generated by the vision encoder. For instance, LLaVA-1.6 (Liu et al., 2024b) can produce 576 to 2880 visual tokens for a single image. The number of visual tokens increases significantly with the addition of more frames. To address this problem, numerous methods have been proposed to reduce the number of visual tokens. One popular direction is to modify the visual resampler that connects the vision encoder and LLM, aiming to extract fewer tokens (Li et al., 2023b;d; Cai et al., 2024; Cheng et al., 2024). Alternative approaches (Chen et al., 2024a; Shang et al., 2024; Jin et al., 2024; Zhou et al., 2024b) employ heuristic techniques to prune or merge the visual features. However, despite these efforts, Table 1 demonstrates that the majority of current LMMs are still limited in their ability to process a large number of frames effectively.

Another issue hindering the development of high-performance long video LMMs is the lack of high-quality long video datasets. In Table 2, we list the average video length of existing video instruction tuning data. Most datasets consist of video clips within 1 minute. Even if some datasets do contain longer videos, the corresponding text pairs are generated by annotating only several frames within that video, lacking long and dense supervision signals.

Given the circumstance, in this paper, instead of reducing the visual tokens, we identify the more critical issue limiting the visual context length in existing LMMs: the context length of the language model backbone. Given a language model, we first extend its context length by training on longer

| Model | Tokens/Frames[*] | Training Max Frames[*] | LM Backbone | LM Context Length |
|---|---|---|---|---|
| MPLUG-Owl-video (Ye et al., 2024) | 256 | 4 | LLaMA | 4K |
| MovieChat (Song et al., 2024b) | 32 | 8 | Vicuna-v0 | 2K |
| Video-LLaVA (Zhang et al., 2023) | 49 | 8 | Vicuna-1.5 | 4K |
| VideoChat (Li et al., 2024b) | 32/196 | 8 | Vicuna-v0 | 2K |
| LLaVA-NeXT-Video (Zhang et al., 2024b) | 144 | 16 | Vicuna-1.5 | 4K |
| ST-LLM (Liu et al., 2024c) | 256 | 16 | Vicuna-1.1 | 2K |
| Video-LLaMA (Cheng et al., 2024) | 32 | 32 | LLaMA-2 | 4K |
| Chat-UniVi (Jin et al., 2023) | 112 | 64 | Vicuna-1.5 | 4K |
| TimeChat (Ren et al., 2024) | 4 | 96 | LLaMA-2 | 4K |
| Video-ChatGPT (Maaz et al., 2023) | 256 | 100 | Vicuna-1.1 | 2K |
| LLaMA-VID (Li et al., 2023d) | 2 | 300 | Vicuna-1.5 | 4K |
| LongVA (Ours) | 144 | - | Qwen2-Extended | 224K+ |

Table 1: To enable longer video inputs, previous works train fewer visual tokens to increase the maximum frames during training. Our LongVA, on the other hand, enables long video capability by extending the backbone language model. [*]We report it based on the best available information from their paper or released codebase.

text data. We then use this context-extended LM as the backbone to perform modality alignment and visual instruction tuning without any long video text pairs. By training this way, the context length of the language model is directly transferred to that of the LMMs. We further proposed *UniRes*, a unified encoding scheme that represents videos as extended images, enhancing the capability fusion between images and videos. To facilitate benchmarking and accurately assess the context length in the visual domain, we created V-NIAH, a synthetic visual benchmark based on the Needle-in-a-haystack test (Gregory, 2024) used in language models. Our model, Long Video Assistant (LongVA), is capable of accurately retrieving visual information from 2000 frames or more than 200K visual tokens. Experiments show that additional frames during inference lead to improved performance on long video question-answering benchmarks, and LongVA achieves state-of-the-art performance among 7B models on the Video-MME (Fu et al., 2024a) and MLVU (Zhou et al., 2024a) dataset. In summary, our paper makes the following contributions:

**(1) Long Context Transfer**: We discovered the *long context transfer* phenomenon where the context of the language model can be directly transferred to the modality-aligned multi-modal models.

**(2) Visual Needle-In-A-Haystack (V-NIAH)**: We proposed the V-NIAH benchmark to test LMMs ability in locating and retrieving visual information over extremely long contexts.

**(3) Long Video Assistant (LongVA)**: With *long context transfer* and *UniRes*, we developed LongVA that can perceive more than 200K visual tokens, achieving SoTA performance on the Video-MME and MLVU dataset.

## 2 RELATED WORK

**Vision Language Connector in Large Multimodal Models**    Existing studies explore different architectures to extract and inject visual features into LLMs. One line of work (Alayrac et al., 2022; Li et al., 2023a; Awadalla et al., 2023; Laurençon et al., 2023), pioneered by Flamingo (Alayrac et al., 2022), adopts a resampler to compress the visual feature and inserts cross-gated attention layers into the LLM. Some other works still use a reampler (Li et al., 2023b; Zhu et al., 2023; Team, 2024b) while directly feeding the image feature into the input layer of the language model. The LLaVA series (Liu et al., 2024b; 2023b;c) use a simple and scalable design to directly project the image features into language model without any pooling or resampling. When the field moves from image-only models to include multi-image and video inputs, more modifications to the visual language connector were proposed. Zhang et al. (2024b) and Cai et al. (2024) use a simple average pooling. Jin et al. (2024) dynamically drop the visual tokens. Cheng et al. (2024) adopt a spatial-temporal convolution to better capture the dynamics of video data and reduce feature size. Our proposed context transfer from text to image is orthogonal to those works and can further enable LMMs to understand more frames.

**Context Extrapolation in Transformer**    Transformer does not directly work on sequences longer than its training length. To alleviate that, various RoPE-based  (Su et al., 2023) extension techniques (Chen et al., 2023a; bloc97, 2024; Rozière et al., 2024; Peng et al., 2023; Ding et al., 2024)

Table 2: Existing Video SFT Datasets

| Dataset Name | Video Length (sec.) | Text Length |
|---|---|---|
| VideoChatGPT-100K (Maaz et al., 2023) | 123.4 | 68.0 |
| LLaVA-Hound-255K (Zhang et al., 2024a) | 52.4 | 37.6 |
| ShareGPT4Video(Chen et al., 2024b) | 26.6 | 273.3 |
| TimeIT (Ren et al., 2024) | 190.8 | 52.5 |
| VideoChat (Li et al., 2024b) | 9.5 | 59.0 |

Table 3: Video Benchmarks

| Benchmark Name | Video Length (sec.) |
|---|---|
| VideoChatGPT (Maaz et al., 2023) | 108.0 |
| NexTQA (Xiao et al., 2021b) | 42.9 |
| EgoSchema (Mangalam et al., 2023) | 179.8 |
| VideoMME (Fu et al., 2024a) | 1017.0 |
| V-NIAH (Ours) | $\infty$ |

have been proposed to allow for training-free context extrapolation. Efforts have also been made on data curation (Fu et al., 2024b; Xiong et al., 2023; Bai et al., 2024) and system optimization (Li et al., 2023c; Liu et al., 2023a; Jacobs et al., 2023) during long context training. There has been limited exploration of the context extrapolation in the domain of LMMs. Liu et al. (2024a) are closest to our work and train LMM with long context language models, but they do not benchmark the effective visual context length of their model.

**Video Language Benchmarks** Recent years have witnessed significant progress in Video Question-AnsweringAntol et al. (2015). To accurately measure the progress of the video LMMs' performance, researchers have developed various benchmarks encompassing a broad spectrum of tasks. These range from fundamental visual perception tasks such as activity recognitionYu et al. (2019a), concept detection (Xu et al., 2017), and counting (Jang et al., 2017), to more complex visual reasoning tasks including compositional (Grunde-McLaughlin et al., 2021), causal (Xiao et al., 2021a; Yi et al., 2019; Xu et al., 2021), and situated reasoning (Wu et al., 2021). However, most of those benchmarks focus on short videos, lacking data and metrics to test LMMs' capability over a long context. Inspired by the NIAH test (Gregory, 2024) in the language model community, we proposed V-NIAH to benchmark LMMs' ability over long visual inputs with the minimum overhead of data collection and human annotation. Several concurrent works also developed multimodal versions of the Needle-in-a-haystack test (Wang et al., 2024c; Zhou et al., 2024a; Song et al., 2024a; Wang et al., 2024a). However, they only measure on several hundreds of frames and lack a strong baseline to properly analyze the properties of visual context length.

# 3 LONG VIDEO ASSISTANT

As in Figure 1, this paper centers around the hypothesis that *if the modality of vision and language can be truly aligned, the capability to handle long contexts could also transfer from text to vision*, and this could happen even without explicit long video training. Our methodology is thus very straightforward. Given a language model, we first perform long context training purely on language to extend its text context (Section 3.1). We then detailed how we augment this language model with long visual capabilities by training solely on short image data in Section 3.2.

## 3.1 TRAINING LONG LANGUAGE MODEL

We use Qwen2-7B-Instruct (Team, 2024c) as the backbone language model and perform continued pretraining with a context length of 224K[1] over a total of 900M tokens. We follow Xiong et al. (2023) to increase RoPE (Su et al., 2023) base frequency during the continued pertaining and specifically set it to 1B. A constant learning rate of 1e-5 is maintained for a batch size of one million tokens across 1,000 training steps. Following Fu et al. (2024b), we construct the dataset used for long context training from Slimpajama (Cerebras, 2023) by upsampling documents longer than 4096 and keeping the domain mixture ratio unchanged. Multiple documents are packed into a single sequence separated by a `BOS` token.

We employed several optimization strategies to perform training on such long sequences. These includes FlashAttention-2 (Dao, 2023), Ring Attention (Liu et al., 2023a; Li et al., 2023c), activation checkpointing, and parameter offload (Rajbhandari et al., 2020). To balance the load across different GPUs, we shard the sequence in a zigzag way (Zhu, 2024) in ring attention. The resulting training framework is memory efficient and maintains very high GPU occupancy. Note that we do not use any

---

[1]224K is the maximum we can fit with 8×A100-80G for Qwen-2-7B. We find that the embedding size significantly impacts the maximum sequence length in our optimized codebase. Qwen2 has a huge vocabulary of 152K tokens. For LLaMA2 with 32K vocabulary, we can train it with 700K context length.

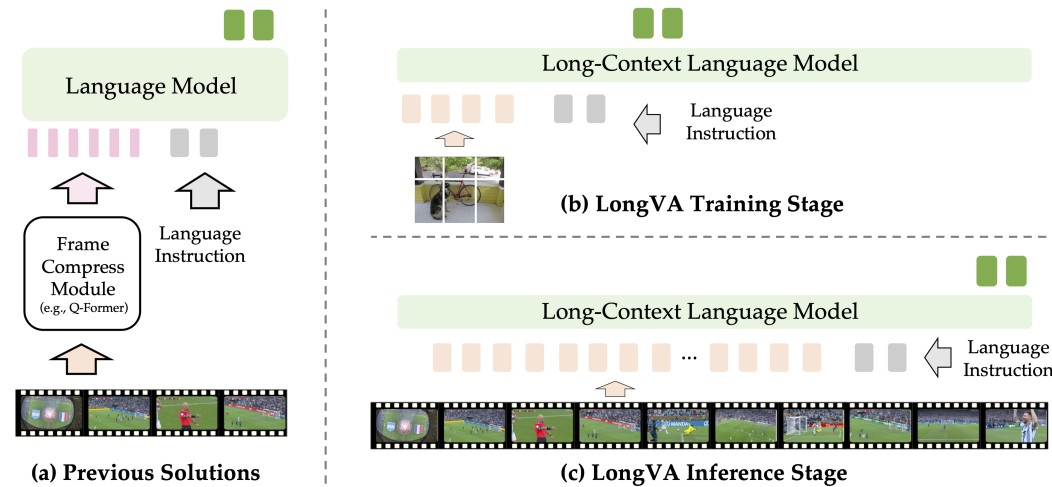

Figure 1: **Left:** to develop long vision models, previous studies proposed better visual resamplers to reduce the number of visual tokens. **Right:** LongVA approaches this problem from the angle of the language model. We leverage image data (short visual input) to align long-context LLM with vision. During the test time, LongVA can zero-shot process extremely long videos, thanks to the property of long context transfer.

parameter-efficient methods such as LoRA (Hu et al., 2021) or approximate attention (Child et al., 2019). With those optimizations, the compute used in long context training is minimal compared to that of language model pretraining, making it feasible for academic budgets. The long context training can finish in 2 days with 8 A100 GPUs.

In Figure 7, we evaluate the extended Qwen2 with the Needle-in-a-haystack (NIAH) test (AI, 2023; Gregory, 2024). It achieves perfect results within the training context length (224K) and generalizes even further. We find the vanilla NIAH to be a relatively trivial benchmark and further test it with 5 distractors randomly inserted into the documents. The detailed configuration can be found in Appendix A.

## 3.2 ALIGNING LONG LANGUAGE MODEL USING SHORT VISION DATA

Inspired by the *AnyRes* encoding scheme in LLaVA-NeXT (Liu et al., 2024b; Li et al., 2024a), we designed *UniRes* that provides a unified encoding scheme for both images and videos, as shown in Figure 2. Unlike *AnyRes* which retains a small base image and flattens ViT patches across the grids, *UniRes* removes the base image, flattens patches within each grid, and 2x2 pool the visual features by default (Appendix B). This approach allows us to maintain consistent representation when extending image data into videos where multiple frames are viewed as multiple grids in a row.

Specifically, *UniRes* divides an input image of resolution $a \times b$ into smaller grids, each with a resolution of $336 \times 336$ pixels. This results in $(a//336) \times (b//336)$ grids. For very high-resolution images, we limit the maximum number of grids to 49, resizing images larger than this threshold. Each grid is separately encoded using `CLIP-ViT-L-336px` (Radford et al., 2021) and then projected through a 2-layer MLP to match the LM's input dimension, resulting in 576 features per grid. We then apply 2x2 average pooling, finally converting an $a \times b$ image into $(a//336) \times (b//336) \times 144$ tokens. During inference, this visual encoding scheme allows videos to be represented as very long images (even though we do not train on videos). An $N$-frame video is treated as an image of size $336 \times (336 \times N)$, divided into $N$ grids where each grid corresponds to a video frame. Using CLIP encoding, MLP projection, and average pooling, an $N$-frame video is encoded into $144N$ visual tokens.

To clearly ablate the long context transfer phenomenon from language to vision, we adopt a *train short, test long* protocol where we only use image-text data during training, but test on long videos. We trained our model using the same data recipe and two-stage training approach as LLaVA-1.6.

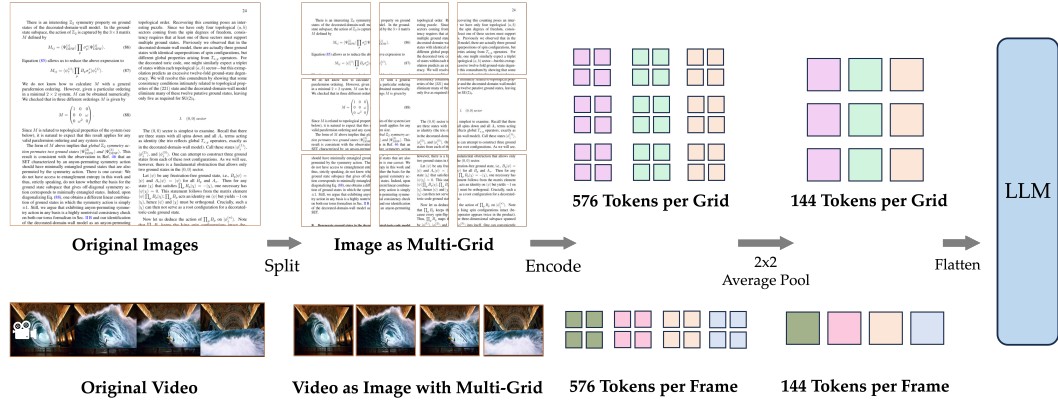

Figure 2: *UniRes*'s unified encoding scheme of images and videos. During training, images are divided into multiple grids. During inference, videos are treated as extended images with each frame considered as a grid.

Our experiments show that compared to *AnyRes*, *UniRes* has slightly lower scores on low-resolution image benchmarks (Table 7) but performs better on V-NIAH (Figure 4) and Video-MME (Table 4). We believe the unified encoding scheme for images and videos is crucial, thus choosing this as the encoding scheme of LongVA. The image-text alignment can be finished in 1.5 days. With 2 days for long context training on text, the total training cost of LongVA is 3.5 days on $8\times$A100-80G.

It is worth noting previous work largely inspired the design choice of LongVA. For example, Xiong et al. (2023) first demonstrates the effectiveness of long context continued pretraining with increased RoPE base frequency (thus decreasing the rotation angles). We sample the long text data following the guidance of (Fu et al., 2024b). We adopt the same vision encoder and training data as that of LLaVA-1.6 (Liu et al., 2024b). We try to keep our methods as simple as possible to clearly show the phenomenon of long context transfer without other confounders.

## 4   V-NIAH

To measure the context length of language models on extremely long input, earlier works calculate perplexity scores over long documents. Recently, many have started using the Needle-in-a-Haystack (NIAH) test to benchmark LLMs' ability to retrieve long context information precisely. We note that there is so far no benchmark to measure the visual context length of LMMs. To evaluate LongVA's capacity to locate and retrieve long-range visual information, we extend the NIAH test from text to video and propose V-NIAH.

As shown in Table 8, we designed 5 video question-answering problems as the needle and inserted each as a single frame into hours-long videos. We sampled the videos at 1 FPS as the visual input. The image of the needle is sourced from existing VQA benchmarks or AI-generated to avoid any contamination. The AI-generated images and questions are purposely chosen to be "counterfactual" or "counter-commonsense", ensuring the model cannot answer based on language knowledge alone. Each question includes a "locating prompt" so that a capable system or human can locate the needle frame from the video haystack and answer the question.

When testing LongVA with visual inputs of up to 3000 frames, one difficulty we encountered was that processing a 200K-token input requires up to 100GB of GPU memory for the KV cache for a 7B LM like LLaMA. Even with advanced LM serving systems like vLLM (Kwon et al., 2023) with tensor parallelism to shard the KV cache across multiple GPUs, the sampling process remains extremely slow due to limited memory and batchsize. To address this, we used "perplexity-based" evaluation to measure the correctness of the model output. We first encode all frames and save their corresponding visual embeddings. During the evaluation, we only load the language model from LongVA and concatenate the visual embeddings, question tokens, and answer tokens for a single forward pass with ring attention. This approach makes the workload compute-bound and eliminates the need to cache the KV state. The model's output is considered correct only if the highest output logits index of all tokens in the answer span matches the correct answer.

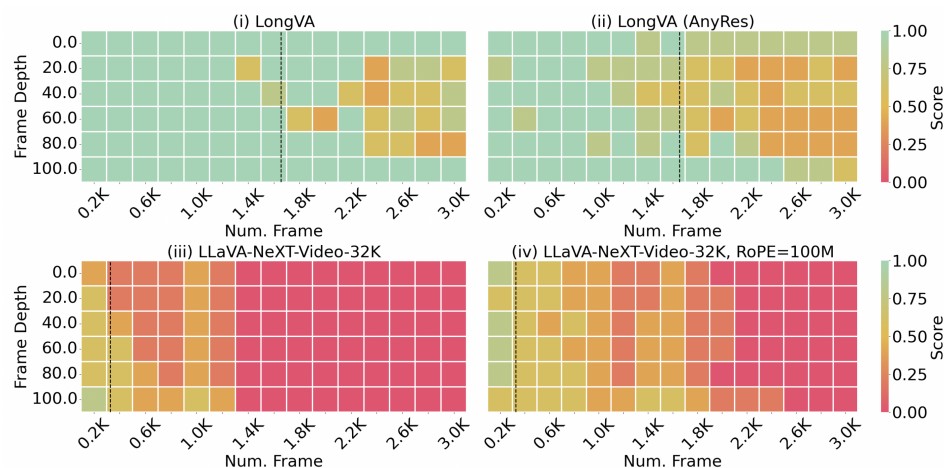

Figure 3: V-NIAH consists of a haystack video, a needle image, and a question related to the needle. The needle is inserted at various positions in the haystack video.

Figure 4: The V-NIAH results of LongVA and its baselines. The x-axis represents the total number of frames in the video haystack. The y-axis shows the position where the needle image is located. For instance, a frame depth of 0% would place the needle image at the very beginning of the video. The black dotted line denotes the training length of the backbone language model, with each frame corresponding to 144 tokens.

## 5  EXPERIMENTS

We primarily assess the long visual capability of LongVA on two benchmarks: V-NIAH (Section 5.1 and Video-MME (Fu et al., 2024a) (Section 5.2). V-NIAH provides quick signals about the visual context length of LongVA. However, it only tests the model's ability to retrieve information and does not cover other abilities necessary for a real-world long video assistant. Therefore, we also include LongVA's performance on Video-MME, a comprehensive evaluation suite for video LMMs that includes diverse data types and qualitative annotations. Video-MME is an ideal benchmark for assessing LMMs' ability to handle long videos in real-world scenarios, given its average video duration of 1017 seconds and the inclusion of short, medium, and long subsets. We further include the benchmark results on MLVU(Zhou et al., 2024a) in Appendix C.

| Model | Qwen2-224K | UniRes |
|---|---|---|
| LLaVA-Next-Qwen2 | × | × |
| LongVA (*AnyRes*) | ✓ | × |
| LongVA | ✓ | ✓ |

Table 5: LongVA and its baselines.

We mainly compare LongVA against other image and video LMMs. To validate the phenomenon of *long context transfer*, we trained LLaVA-Next-Qwen2, a baseline model based on Qwen2-7B-Instruct using the LLaVA-NeXT (Liu et al., 2023b; Li et al., 2024a) training recipe. Additionally, we trained LongVA (*AnyRes*) to showcase the advantages of our *UniRes* encoding scheme. The difference between LongVA and our baselines can be found in Table 5.

| Model | LLM Params | Frames | Short | Medium | Long | Overall |
|---|---|---|---|---|---|---|
| InternVL-Chat-V1.5 (Chen et al., 2023b) | 20B | 10 | 60.2 | 46.4 | 45.6 | 50.7 |
| LLaVA-NeXT-Video-34B (Zhang et al., 2024b) | 34B | 32 | 61.7 | 50.1 | 44.3 | 52.0 |
| VILA-1.5 (Lin et al., 2023b) | 34B | 8 | 68.1 | 58.1 | 50.8 | 59.0 |
| Qwen-VL-Chat (Team, 2024b) | 7B | 4 | 46.9 | 38.7 | 37.8 | 41.1 |
| Video-LLaVA (Lin et al., 2023a) | 7B | 8 | 45.3 | 38.0 | 36.2 | 39.9 |
| ST-LLM (Liu et al., 2024c) | 7B | 64 | 45.7 | 36.8 | 31.3 | 37.9 |
| VideoChat2-Mistral (Li et al., 2024b) | 7B | 16 | 48.3 | 37.0 | 33.2 | 39.5 |
| Chat-UniVi-V1.5 (Jin et al., 2023) | 7B | 64 | 45.7 | 40.3 | 35.8 | 40.6 |
| VideoLLaMA2 (Cheng et al., 2024) | 8B | 16 | 56.0 | 45.4 | 42.1 | 47.9 |
| LLaVA-NeXT-Qwen2 | 7B | 32 | 59.8 | 48.2 | 44.7 | 50.9 |
| LongVA | 7B | 8 | 55.6 | 46.0 | 41.7 | 47.7 |
| | | 16 | 59.9 | 47.0 | 43.8 | 50.2 |
| | | 32 | 61.7 | 49.1 | 45.9 | 52.2 |
| | | 64 | **61.8** | **51.7** | 44.6 | 52.7 |
| | | 128 | 61.6 | 50.6 | **47.1** | **53.1** |
| | | 384 | 60.9 | 49.9 | 46.1 | 52.3 |

Table 4: Performance comparison of various LMMs on Video-MME (Fu et al., 2024a) *without subtitles*. LongVA achieves state-of-the-art results among 7B models. Its performance also increases with denser sampling of video frames.

## 5.1 V-NIAH RESULTS

**Long context transfers from language to vision** Figure 4 shows the V-NIAH performance of LongVA and other LMMs. Specifically, Figure 4 (iii) demonstrates that the visual context length of LLaVA-NeXT-Video-32K (Zhang et al., 2024b) is constrained by the 32K context length of its language backbone, Mistral-7B-Instruct-v0.2 (Jiang et al., 2023), equivalent to approximately 200 frames. Beyond this limit, the V-NIAH accuracy drops significantly. As a stronger baseline, we include the results of LLaVA-NeXT-Video-32K enhanced with a training-free length extrapolation algorithm (bloc97, 2024) by increasing its RoPE base frequency. We empirically determine the optimal extrapolation frequency by choosing from [3M, 10M, 30M, 100M, 300M, 1B]. As indicated in Figure 4 (iv), although this training-free extrapolation allows the model to process information across an extended context, the improvement is marginal. These findings led us to develop LongVA, a model that unlocks the visual context by extending the language model purely on text. As shown in Figure 4 (i), LongVA can almost perfectly retrieve information and answer the needle question for input frames fewer than 2000. Although we only trained LongVA's language backbone on a context length of 224K (equivalent to 1555 frames), it generalizes well beyond that, maintaining satisfactory performance within 3000 frames. Those results clearly corroborate of hypothesis of *long context transfer*.

**Unified encoding enables better visual context extrapolation** We also present the V-NIAH heatmap of LongVA trained with *AnyRes* encoding scheme, keeping all other factors unchanged in Figure 4 (ii). LongVA-*AnyRes* demonstrates strong retrieval capabilities. However, its performance still lags behind LongVA trained with UniRes. We believe that the unified representation of images and videos in UniRes, where a video is encoded in the same way as a long image, enhances the long context transfer from language to vision. This approach also facilitates effective training with short vision data (images) and enables zero-shot understanding of long videos during inference.

## 5.2 VIDEO EVALUATION

On Video-MME (Table 4), LongVA achieves *state-of-the-art* performance among LMMs under 10B parameters, rivaling much larger ones such as LLaVA-NeXT-Video-34B (Zhang et al., 2024b) and InternVL-Chat-V1.5 (Chen et al., 2023b). Notably, LongVA is trained without any video data, so its performance on video can be considered *zero-shot*. As the number of sampled frames increases, LongVA shows improved performance on the long subset, handling up to 384 frames[2]. Even though LongVA's score slightly drops when we upsample from 128 to 384 frames, it maintains a competitive performance. To our knowledge, LongVA is the *only* open-source model that can handle such large input frames on Video-MME. These findings highlight the *long context transfer* effect, where

---

[2]We limited our analysis to 384 frames due to computational and memory constraints as detailed in Section 4.

| Model | frames | NeXTQA | | ActivityNetQA | VideoChatGPT | | | | | Video-DD |
| | | MC | OE | Score | Consistency | Correctness | Detail | Context | Temporal | Score |
|---|---|---|---|---|---|---|---|---|---|---|
| LLaVA-NeXT-Video | 32 | 57.93 | 26.90 | 3.20 | 3.12 | 3.39 | 3.29 | 3.92 | 2.60 | 3.32 |
| LongVA | 8 | 50.78 | 27.71 | 2.73 | 3.73 | 3.09 | 3.14 | 3.72 | 2.39 | 3.19 |
| LongVA | 16 | 61.61 | 27.87 | 2.78 | 3.61 | 3.13 | 3.15 | 3.75 | 2.40 | 3.22 |
| LongVA | 32 | 67.08 | 27.87 | 2.80 | 3.65 | 3.08 | 3.10 | 3.74 | 2.28 | 3.19 |
| LongVA | 64 | 68.27 | 27.81 | 2.84 | 3.64 | 3.05 | 3.09 | 3.77 | 2.44 | 3.14 |
| LongVA-DPO | 32 | 69.26 | 28.02 | 2.80 | 4.07 | 3.55 | 3.32 | 4.09 | 2.86 | 3.58 |

Table 6: Video evaluation results for LongVA on various short video benchmarks with comparison to 7B scale models.

LongVA, originating from a long context language model, can process significantly more frames than its baseline, despite being trained on the same multimodal data.

We also tested LongVA on shorter benchmarks with average video durations under 120 seconds. As indicated in Table 6, although LongVA scores higher with more densely sampled frames on datasets such as NeXTQA (Xiao et al., 2021b) and ActivityNetQA (Yu et al., 2019b), the gains quickly plateau and are not as significant as those observed in Video-MME, which can be attributed to the shorter duration of these datasets. On the VideoChatGPT (Maaz et al., 2023) and Video Detailed Description (Video-DD) (Maaz et al., 2023) benchmarks, increasing frames does not lead to better performance, and LongVA generally achieves lower scores compared to LLaVA-NeXT-Video-7B. Since both benchmarks use OpenAI's GPT API as a judge, we believe their metrics are closely related to the answering format. To address this, we perform a lightweight Direct Preference Optimization (DPO) on the LLaVA-Hound-DPO (Zhang et al., 2024a) dataset. We observe significantly improved performance for LongVA-DPO, confirming the findings in Zhang et al. (2024a).

## 5.3 IMAGE EVALUATION

| Model | AI2D | ChartQA | DocVQA | InfoVQA | RealworldQA | MMMU |
|---|---|---|---|---|---|---|
| LLaVA-1.6-Vicuna | 66.6 | 54.8 | 74.4 | 37.1 | 57.8 | 35.1 |
| LLaVA-NeXT-LLaMA3 | 71.6 | 69.5 | 78.2 | 37.6 | 60.0 | 41.7 |
| LLaVA-NeXT-Qwen2 | 73.5 | 74.0 | 81.3 | 42.0 | 61.6 | 41.9 |
| LongVA (*AnyRes*) | 73.1 | 74.4 | 81.5 | 43.3 | 62.4 | 42.1 |
| LongVA (*UniRes*) | 70.7 | 70.4 | 80.8 | 49.4 | 60.0 | 42.6 |

Table 7: Image evaluation results for LongVA on multiple benchmarks. Compared to other image multimodal models, our methods maintain high performance and achieve better scores on InfoVQA(Mathew et al., 2020).

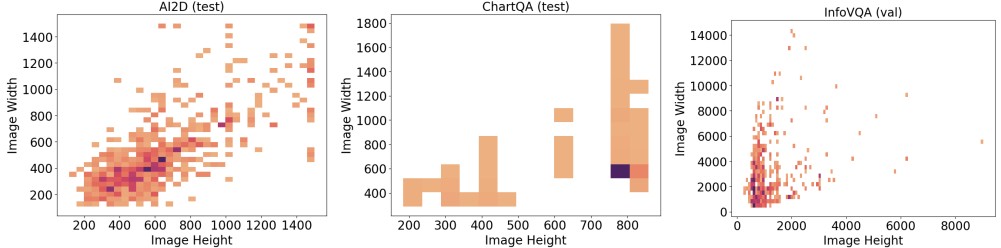

Figure 5: The 2D-histogram of the image width and height of different image benchmarks. InfoVQA(Mathew et al., 2020) consists of many high-resolution images compared to other benchmarks.

We further evaluate our model on various image benchmarks to investigate the image performance of LongVA (Table 7). Compared to the LongVA (*AnyRes*) baseline, LongVA with *UniRes* achieves significantly increased performance on InfoVQA (Mathew et al., 2020), while the scores drop to some extent on AI2D (Kembhavi et al., 2016) and ChartQA (Masry et al., 2022). On DocVQA (Mathew et al., 2020), RealworldQA (xAI, 2024), and MMMU (Yue et al., 2023), LongVA is able to match its baseline's performance. To better understand this phenomenon, we recorded and analyzed the image size of those datasets, as shown in Figure 5. We found that InfoVQA consists of higher-resolution images, while many images in AI2D and ChartQA are smaller than 768×768. Compared to *Anyres*,

*UniRes* operate $2\times2$ average pooling on each image, reducing to $1/4$ visual tokens per image grid. However, the grid upper bound is set to 49 for *UniRes* while 4 for *AnyRes*, so *UniRes* may produce more image grids if the input images are of higher resolution. By using more grids per image, *UniRes* allocates more visual tokens on datasets such as InfoVQA, achieving superior performance compared to the previous 7B LLaVA model. However, most of the images in ChartQA and AI2D require fewer than 4 grids to represent. This may explain why the image performance decreases on those benchmarks.

# 6 QUALITATIVE RESULTS

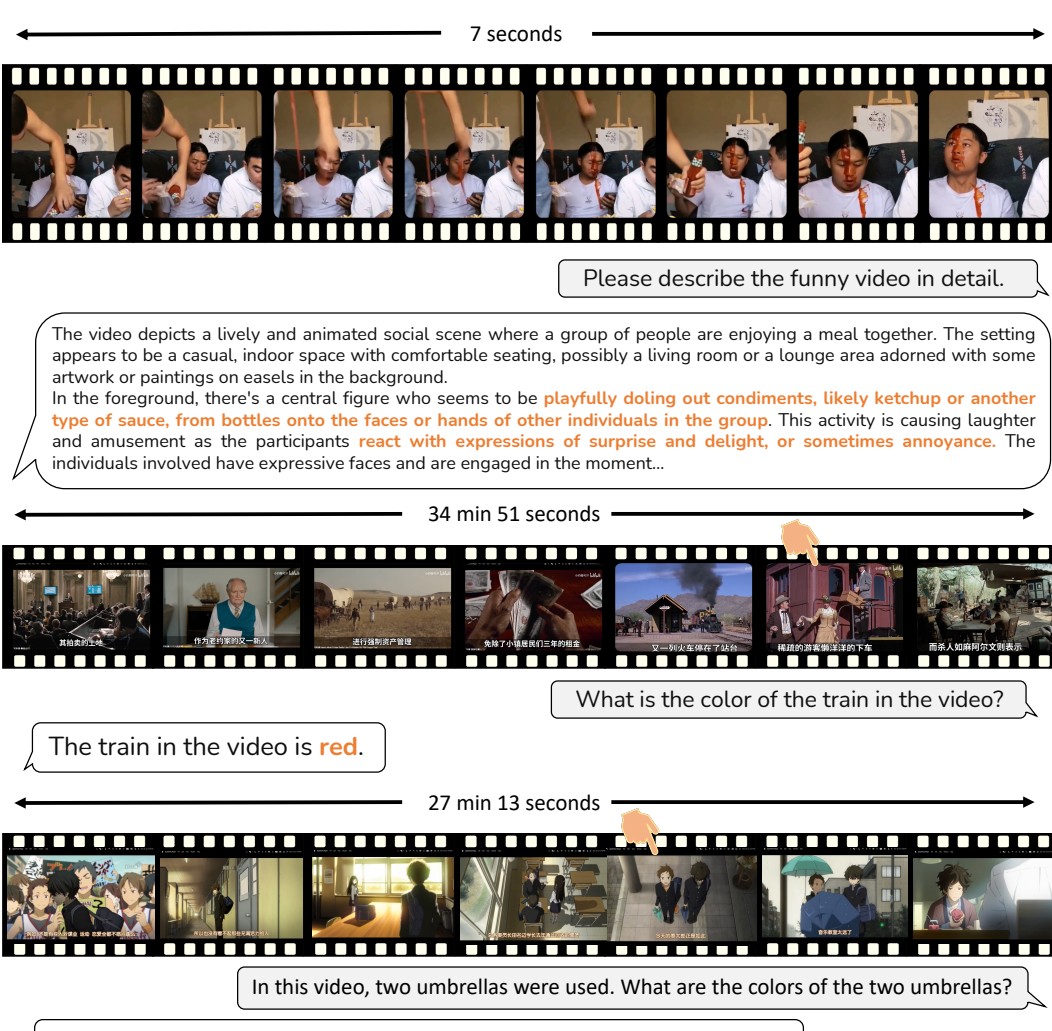

Figure 6: Qualitative Result of the LongVA-DPO. Our model demonstrates superior capability in understanding both short and long videos.

The qualitative results of LongVA-DPO are illustrated in Figure 6. The short video example comes from (Xie et al., 2023) and the two long videos are sourced from link1 and link2, respectively. In the figure, LongVA accurately describes the short, humorous video involving individuals playfully interacting with condiments. It also identifies specific details in long videos, such as the color of a train and the colors of umbrellas used in a scene, showcasing its proficiency in retrieving and interpreting visual information over extended video contexts. These capabilities highlight LongVA's potential to overcome the challenges associated with processing and understanding extremely long videos.

# 7 VISUAL NEEDLE IN A HAYSTACK TEST

Table 8 lists the five VQA needles we used for V-NIAH. The 5 visual questions and answers are the only places where human annotation is involved in the construction of V-NIAH, making it an ideal testbed to benchmark LMMs' long context capability.

---

**V-NIAH Needles**

---

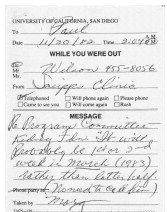

**Question:** Find the frame of the 'While You Were Out' note. What is the name of the university on that note?
A. University of California, Los Angeles
B. University of California, San Diego
C. University of California, Berkeley
D. University of California, Santa Barbara
Answer with the option's letter from the given choices directly.

**Answer:** B

---

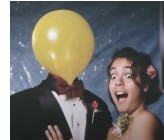

**Question:** Find the frame of a couple in a wedding. Inside the frame, there is a balloon on the bridegroom's head. What is the color of that balloon?
Answer the question using a single word or phrase.

**Answer:** Yellow

---

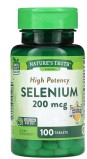

**Question:** Find the frame with the image of Selenium tablets. How many mg does each tablet contain?
Answer the question using a single word or phrase.

**Answer:** 200

---

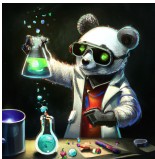

**Question:** Find the frame of a scientist. The scientist is a...
A. Bird
B. Elephant
C. Panda
D. Dog
Answer with the option's letter from the given choices directly.

**Answer:** C

---

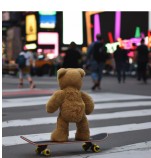

**Question:** Find the frame of a teddy bear. Where is this teddy bear?
A. Times Square
B. Eiffel Tower
C. Taj Mahal
D. Sydney Opera House
Answer with the option's letter from the given choices directly.

**Answer:** A

---

Table 8: The design of the 5 visual question-answering problems used as the needle in V-NIAH.

# 8 CONCLUSION

This work addresses the challenges of understanding long videos in Large Multimodal Models. By extending the language model on text and then aligning this extended model with visual inputs, we significantly improved the capability of LMMs to handle long videos thanks to the *long context transfer* phenomenon. Our model, LongVA, shows improved performance with more input frames and achieves state-of-the-art results on Video-MME. Additionally, we introduce a synthetic benchmark, V-NIAH, to effectively measure the visual context length of video LMMs. We hope this work inspires further research in the field of long video LMMs and multimodal agents.

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

# APPENDIX

## A NEEDLE IN A HAYSTACK TEST

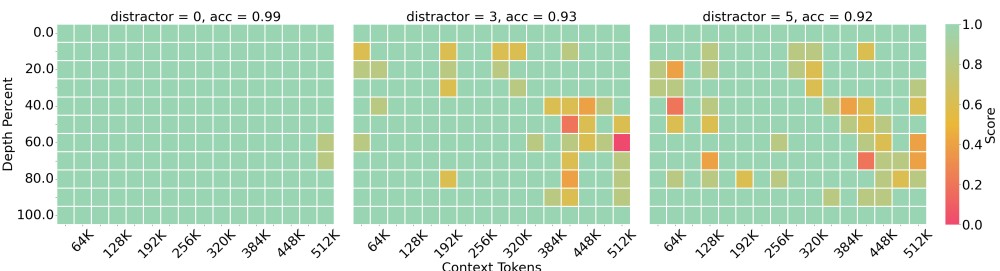

Figure 7: The NIAH results of Qwen-7B-Instruct after long context training.

When evaluating the Needle In A Haystack task (Gregory, 2024), we focus specifically on an easier-to-evaluate variant (AI, 2023) that involves identifying and retrieving random numbers associated with various randomly assigned cities from the context. The input to the language model has below template:

```
This is a very long story book: <book> {haystack + needle + haystack} </
    book>.\n Based on the content of the book, Question: What is the
    special magic Singapore number? Answer: The special magic Singapore
    number is:
```

We insert a needle with the key Singapore and a 7-digit randomly sampled magic number as the value into the haystack of Paul Graham's Essays. The needle has the following format:

```
\nThe special magic {City} number is: {XXXXXXX}.\n
```

We iterate over various document depths (where the needle is placed) and context lengths to measure the performance. For each depth and context length, we conducted the test 5 times, each time with a different 7-digit needle. We also come up with a harder version where we also insert several (3 or 5) other needles with the same format but different city name as distractors. The results are shown in Figure 7.

## B UNIRES ENCODING SCHEME

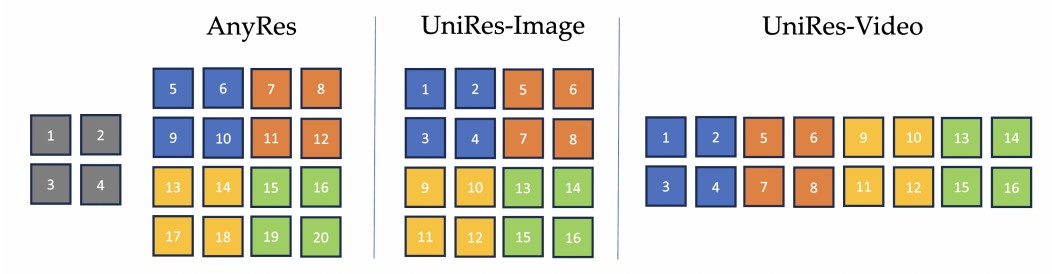

Figure 8: The difference between *AnyRes* and *UniRes*, assuming the image is divided into 2x2 grids and the video has 4 frames. The number indicates the flattening order. Additionally, *UniRes* applies 2x2 average pooling to both images and videos *after* the MLP projector between the vision encoder and the language model.

Figure 8 indicates the difference between *AnyRes* and *UniRes*. Given a high-resolution image and assuming we use `CLIP-ViT-L-336px` as the vision encoder, both *AnyRes* and *UniRes* will divide

it into multiple grids, each with the size 336x336. However, *AnyRes* will have a smaller version of the full image as the base image and prepended before the high-resolution image grids. Additionally, *UniRes* flattens the encoded image feature in a raster-order *within* each grid, while *AnyRes* combines all the grids as a big feature map and flattens them *across* the border of the grid. *UniRes* also apply 2x2 average pooling on the image feature. As shown in the rightmost part of Figure 8, the design of *UniRes* allows us to unifiedly encode videos as well. A video is treated as an extended image where each frame is considered as an image grid.

## C   MLVU RESULTS

| Methods | Input | Holistic | | | Single Detail | | Multi Detail | | | | Averages | |
|---|---|---|---|---|---|---|---|---|---|---|---|---|
| | | TR | AR | VS | NQA | ER | PQA | SSC | AO | AC | M-Avg | G-Avg |
| GPT-4o (OpenAI, 2024) | 0.5 fps | 87.4 | 74.5 | 4.90 | 64.8 | 57.1 | 65.1 | 6.69 | 56.7 | 46.3 | 64.6 | 5.80 |
| LLaVA-1.6 (Liu et al., 2024b) | 16 frm | 60.6 | 41.0 | 2.11 | 43.1 | 38.4 | 41.0 | 4.35 | 25.5 | 25.7 | 39.3 | 3.23 |
| InternVL-1.5 (Chen et al., 2023b) | 16 frm | 78.8 | **67.0** | 3.16 | 52.7 | 43.5 | 54.4 | 4.88 | 32.8 | 23.8 | 50.4 | 4.02 |
| MovieChat (Song et al., 2024b) | 2048 frm | 29.5 | 25.0 | 2.33 | 24.2 | 24.7 | 25.8 | 3.23 | 28.6 | 22.8 | 25.8 | 2.78 |
| TimeChat (Ren et al., 2024) | 96 frm | 23.1 | 27.0 | 2.54 | 24.5 | 28.4 | 25.8 | 4.29 | 24.7 | **32.0** | 30.9 | 3.42 |
| LLaMA-VID (Li et al., 2023d) | 1 fps | 50.8 | 34.5 | 3.22 | 30.1 | 32.7 | 32.5 | 5.22 | 23.9 | 27.8 | 33.2 | 4.22 |
| MA-LMM (He et al., 2024) | 1000 frm | 51.9 | 35.5 | 2.12 | 43.1 | 38.9 | 35.8 | 4.80 | 25.1 | 24.3 | 36.4 | 3.46 |
| ShareGPT4Video (Chen et al., 2024b) | 16 frm | 75.8 | 51.5 | 2.52 | 47.6 | 43.2 | 48.4 | 5.02 | 34.0 | 23.3 | 46.4 | 3.77 |
| VideoChat2_HD (Li et al., 2024b) | 16 frm | 77.3 | 60.5 | 3.38 | 46.2 | 48.9 | 50.1 | 4.59 | 23.2 | 29.1 | 47.9 | 3.99 |
| VideoLlaMA2 (Cheng et al., 2024) | 16 frm | 74.6 | 64.5 | 2.79 | 49.9 | 43.8 | 45.1 | 5.18 | 34.0 | 27.4 | 48.5 | 3.99 |
| LongVA (ours) | 256 frm | **83.3** | 58.5 | **3.39** | **69.3** | **50.0** | **67.2** | **5.26** | **38.6** | 27.2 | **56.3** | **4.33** |

Table 9: Evaluation results by the authors of MLVU (Zhou et al., 2024a). The highest scores excluding GPT-4o are bolden. TR: Topic Reasoning. AR: Anomaly Recognition. VS: Video Summary; NQA: Needle QA; ER: Ego Reasoning; PQA: Plot QA; SSC: Sub-Scene Captioning; AO: Action Order; AC: Action Count; M-Avg: the average performance of multiple-choice tasks; G-Avg: the average performance of generation tasks.

Table 9 includes the evaluation results by the authors of MLVU (Zhou et al., 2024a) on their benchmark. LongVA achieves *state-of-the-art* results among open-source models and is only second to GPT-4o.

