# OpenReview forum: "Long Context Transfer from Language to Vision"
_ICLR.cc/2025/Conference — Submitted to ICLR 2025_

### Official Review · Reviewer_DYq4 · 2024-10-27

**Soundness:** 4
**Presentation:** 3
**Contribution:** 3
**Rating:** 5
**Confidence:** 5

**Summary:**

This paper addresses the challenge of enabling Large Multimodal Models (LMMs) to process and understand extremely long videos. While previous works have focused on reducing the number of visual tokens through visual resamplers, this paper approaches the problem from the perspective of the language model. By extending the context length of the language backbone, the authors enable LMMs to process significantly more visual tokens without any additional video training. They term this phenomenon "long context transfer," where the extended context capabilities of the language model transfer to the vision modality.

To evaluate this capability, the authors introduce V-NIAH (Visual Needle-In-A-Haystack), a synthetic benchmark inspired by the NIAH test for language models, designed to test LMMs' ability to locate and retrieve information from extremely long visual contexts. They propose LongVA (Long Video Assistant), an LMM that, through context length extrapolation in the language model, can process over 200K visual tokens or 2000 frames without additional complexities or long video training. LongVA achieves state-of-the-art performance among 7B-scale models on the Video-MME and MLVU datasets by densely sampling more input frames.

**Strengths:**

Novel Approach: The paper introduces a novel method for enabling LMMs to process extremely long videos by extending the context length of the language model, rather than reducing the number of visual tokens. This shift in perspective is innovative and opens new avenues for research in multimodal learning.

Long Context Transfer Phenomenon: The discovery of the long context transfer phenomenon is significant. It demonstrates that the extended context capabilities of a language model can directly benefit the vision modality without additional long video training, which is a valuable insight.

V-NIAH Benchmark: The development of V-NIAH provides a useful synthetic benchmark for evaluating LMMs' ability to handle long visual contexts. This can serve as a valuable tool for the research community to assess and compare models.

Strong Experimental Results: LongVA achieves state-of-the-art performance among 7B-scale models on the Video-MME and MLVU datasets. The experimental results are comprehensive and support the claims made in the paper.

Ablation Studies: The paper includes thorough ablation studies that analyze the properties of long context transfer and the impact of different components of the model, adding depth to the evaluation.

**Weaknesses:**

Latest LLMs like llama3.2 already has context length up to 128k, which makes this work less valuable.

Limited Real-World Evaluation: While V-NIAH is a useful synthetic benchmark, it may not fully capture the complexities of real-world long video understanding. The reliance on synthetic data could limit the assessment of the model's practical applicability in real-world scenarios.

Potential Limitations: The paper could delve deeper into potential limitations or failure cases of the long context transfer phenomenon. Understanding scenarios where this approach might not be effective would strengthen the work.

Clarity in Presentation: While the paper is generally well-written, some sections could benefit from clearer explanations. For example, more detailed descriptions of technical aspects and improved figures could enhance overall readability.

**Questions:**

Limitations of Long Context Transfer: Can the authors provide more insights into the limitations of the long context transfer phenomenon? Are there scenarios where this approach might not be effective?

---

> ### Author Response · Authors · 2024-11-21
>
> We appreciate reviewer DYq4’s valuable suggestions. We have addressed them in detail in the following response.
>
> > [W1] Less valuable due to llama 3.2.
>
> We appreciate the reviewer’s feedback, but we would like to clarify that the primary contribution of our work is not simply proposing SoTA model. Instead, our focus is on uncovering, analyzing, and rigorously documenting the long context transfer phenomenon and introducing a synthetic benchmark for long video understanding. These contributions are foundational and intended to inspire future advancements in this domain.
>
> Regarding LLaMA 3.2, it is important to note that this model—released after the completion of our work—was not available during the development of LongVA. We actually believe the release of llama 3.2 is a good thing. We advocate for a standard practice to either extend the language model or directly use the long context language model for training large multimodal models.
>
> > [W2]  Reliance on synthetic benchmark.
>
> We appreciate the reviewer’s concern and would like to clarify that V-NIAH is not the sole benchmark used in our evaluation. In addition to V-NIAH, we have conducted extensive evaluations on real-world datasets such as Video-MME and MLVU.
> The key advantage of V-NIAH lies in its scalability—it can accommodate arbitrarily long videos without the need to recollect or reannotate benchmark datasets as video lengths increase. This makes V-NIAH an efficient and effective tool for quickly assessing a model’s visual context length capabilities. By combining V-NIAH with evaluations on real-world datasets, we aim to provide a balanced and comprehensive assessment. The two approaches complement each other.
>
> > [W3]  The paper could delve deeper into potential limitations or failure cases of the long context transfer phenomenon.
>
> We acknowledge the importance of discussing the potential limitations of the long-context transfer phenomenon. We offer some below and will add a limitation section in the updated manuscript:
>
> Long context transfer is not unlimited. LongVA’s performance saturates when we increase the input frames to 384 on VideoMME.

---

> ### Author Response · Authors · 2024-11-27
>
> Dear reviewer DYq4,
>
> We sincerely appreciate your feedback. We take every comment seriously and hope our response can address your concerns. If you have any further questions, we’d be more than happy to respond.

---

### Official Review · Reviewer_kGWU · 2024-11-03

**Soundness:** 3
**Presentation:** 3
**Contribution:** 2
**Rating:** 5
**Confidence:** 4

**Summary:**

The paper addresses a significant limitation in large multimodal models (LMMs) concerning their ability to process extremely long video sequences. The existing approaches primarily focused on reducing the number of visual tokens using visual resamplers, this work proposes a novel perspective by extending the context length of the language model backbone, which allows LMMs to interpret a substantially larger number of visual tokens—up to 2000 frames or over 200,000 visual tokens—without requiring additional video training.  The authors also introduce V-NIAH, a synthetic benchmark designed to assess LMMs' generalization capabilities in handling long visual contexts. The results indicate that the proposed method achieves state-of-the-art performance on the Video-MME task among 7B-scale models.

**Strengths:**

* The paper is well-written and easy to understand. The figure illustration and captions are informative.
* The authors contributed the V-NIAH benchmark, which is an effective benchmark to test LMMs' ability to locate and retrieve visual information over extremely long contexts.
*  Comprehensive evaluations on two video understanding benchmarks (Video-MME and MLVU) beat most of the baselines.

**Weaknesses:**

* The technical contributions of this paper are limited. It appears that the authors' only contribution is the benchmark used to evaluate long-context understanding capabilities, which may not be sufficient for a paper presented at the main conference of ICLR.
* The authors' design to enhance the capability of video understanding in long contexts only involves using a large language model (LLM) with long-context capabilities. Such an improvement is obvious and does not provide new insights.
* Where is the **transfer** mentioned in the title of the article reflected? The authors do not seem to have made any special design considerations for transfer, aside from using a long-context LLM as the base model.

**Questions:**

Refer to weakness.

---

> ### Author Response · Authors · 2024-11-21
>
> We thank reviewer kGWU for the comments!
>
> > [W1] Limited contribution
>
> We believe the key contribution of our paper is in demonstrating and analyzing the transfer of long-context understanding capabilities from text to video. This is a novel and important observation.
>
> Reviewer 9tBS and Reviewer 92h6 have raised concerns about how long video understanding can emerge from training solely on long text and image-text data. This highlights that the phenomenon of long-context transfer is not obvious and may even seem counterintuitive. This makes our findings both significant and worth further exploration.
>
> Through leveraging long-context transfer and conducting comprehensive experiments, we show that LongVA achieves strong performance in long-context video understanding. This is achieved with lightweight training on single-image datasets.
>
> > [W2] Long context language model, obvious improvements.
>
> We respectfully disagree with the assertion that using a long-context language model to enhance video understanding is "obvious." Concurrent work (Wu et al.) highlights that existing open-source models struggle to process videos longer than 64 frames effectively.
>
> In contrast, LongVA is the first open-source model (outside of proprietary efforts from companies like Google and OpenAI) to show consistent performance improvements with increasing input frames, maintaining robust accuracy even at 384 frames. We run more experiments on VideoMME of other open-source LMMs:
>
> | Model      | Frames | Short  | Medium | Long  | Total  |
> |------------|--------|--------|--------|-------|--------|
> | LongVA     | 128    | 61.1   | 50.4   | 46.2  | 52.6   |
> | LongVA     | 64     | 61.4   | 50.9   | 45.0  | 52.4   |
> | LongVA     | 32     | 61.1   | 45.4   | 45.4  | 51.8   |
> | IDEFICS2   | 64     | 16.9   | 15.4   | 12.6  | 15.0   |
> | IDEFICS2   | 32     | 55.9   | 49.6   | 42.7  | 49.37  |
>
> **Table: VideoMME Results of Open-Source LMMs**
>
> This ability to achieve state-of-the-art results with no video data during training represents a non-obvious and impactful contribution to the field. We hope this clarification helps to better contextualize the novelty and significance of our work.
>
> H. Wu, D. Li, B. Chen, and J. Li, “Longvideobench: A benchmark for long-context interleaved video-language understanding,” arXiv preprint arXiv:2407.15754, 2024.
>
> > [W3] Where is the **transfer** mentioned in the title of the article reflected?
>
> We acknowledge the importance of clearly articulating how the "transfer" in the title is reflected in our work. The core contribution lies in demonstrating **modality transfer** from long-text and image training to video understanding. We intentionally do not include: 1. Short or long video data. 2. Multi-image data in the training process of LongVA. Thus, the long video understanding capability is zero-shot transferred from the long text capability and image understanding capability.
>
> The success of this transfer is demonstrated through extensive experiments, as presented in Table 4 and Figure 4. These results highlight how LongVA leverages its long-text and image training to achieve competitive performance in long video understanding tasks without requiring extensive video-specific training.

---

> > ### Comment · Reviewer_kGWU · 2024-11-25
> >
> > Thanks for the efforts made by the authors in the rebuttal. After reading the response and the comments from other reviewers, I still think the technique novelty of this paper is not sufficient to be accepted. The contribution of simply using large language models with long-context capabilities to improve long-context video understanding is marginal.

---

### Official Review · Reviewer_yaCh · 2024-11-04

**Soundness:** 4
**Presentation:** 4
**Contribution:** 3
**Rating:** 8
**Confidence:** 4

**Summary:**

This paper focuses on the challenge of processing long video sequences in Large Multimodal Models (LMMs). Instead of focusing on reducing visual tokens, the authors identify the language model's context length as the key limitation and propose a novel approach called "long context transfer." The method first extends the context length of the language model through training on longer text data, then performs modality alignment without requiring long video-text pairs. They introduce UniRes, a unified encoding scheme representing videos as extended images. They propose V-NIAH, a synthetic benchmark based on the Needle-in-a-haystack test, to evaluate visual context length capabilities. Their resulting model, Long Video Assistant (LongVA), can effectively process over 200K visual tokens (approximately 2000 frames) and achieves state-of-the-art performance on Video-MME and MLVU datasets among 7B-parameter models, demonstrating that increased frame input during inference leads to improved performance on long video question-answering tasks.

**Strengths:**

1. Novel Perspective: The paper presents a fresh perspective on handling long video sequences in LMMs by identifying the language model's context length as the primary bottleneck rather than following the conventional approach of reducing visual tokens. This reframing of the problem leads to a novel solution pathway.

2. Technical Innovation: The discovery of the "long context transfer" phenomenon represents a significant contribution to the field. The ability to transfer extended context capabilities from language to vision without explicit long video-text pair training is both elegant and practical, potentially influencing future research directions in multimodal learning.

3. Benchmark Contribution: The introduction of V-NIAH provides the community with a much-needed evaluation framework for assessing visual context length capabilities in LMMs. This standardized benchmark addresses a gap in existing evaluation metrics and will facilitate future research in long-form video understanding.

4. Practical Impact: The achieved results are impressive, with LongVA demonstrating the ability to process over 200K visual tokens and showing state-of-the-art performance on established benchmarks. The scalability of the approach (showing improved performance with more input frames) suggests real-world applicability.

5. Methodological Clarity: The proposed UniRes encoding scheme offers a unified approach to handling both images and videos, promoting better integration of these modalities and potentially simplifying the architecture of multimodal systems.

**Weaknesses:**

1. Visual Encoding Ablations: The paper lacks comprehensive analysis of different visual encoding schemes' impact on performance. Specifically:
   - The comparison between UniRes, AnyRes, and Higher-AnyRes is not thoroughly explored
   - The effect of including or excluding base image encoding is not clearly demonstrated
   - The impact of different layout configurations for extended images (1 x N, N x 1, etc.) is not investigated
   These ablation studies would provide valuable insights into the robustness and generalizability of the proposed approach.

2. Limited Context Analysis: Given that the paper's core contribution is the long context transfer phenomenon, a more detailed analysis of how different context lengths in the language model correlate with visual understanding capabilities would strengthen the claims. The mechanism behind this transfer phenomenon could be better explained.

3. Typos: The paper contains several typographical errors, specifically in the references on lines 123 and 125, where parentheses and spaces are missing. While minor, such oversights affect the paper's professional presentation and should be addressed in the final version.

**Questions:**

See "Weakness" 1 & 2.

---

> ### Author Response · Authors · 2024-11-21
>
> We thank reviewer yaCh for the insightful comments!  Below, we address the reviewer's concerns in turn.
>
> > [W1] Visual Encoding Ablations
>
> We compare UniRes and AnyRes in Figure 4. UniRes demonstrates significantly better length extrapolation capabilities for long videos than AnyRes. We further run LongVA (AnyRes) on VideoMME and compare it with LongVA (UniRes). Please refer to our general response for the detailed results.
>
> In Table 7, we observe that AnyRes outperforms UniRes on certain benchmarks, such as AI2D and ChartQA. This may be because UniRes does not include a base image feature and applies 2×2 pooling by default, resulting in fewer visual tokens per image. However, on InfoVQA—a high-resolution benchmark requiring detailed image understanding—UniRes outperforms AnyRes by a large margin. This result further confirms the exceptional long-context extrapolation capability of the UniRes encoding scheme.
>
> > [W2] Limited context analysis.
>
> We further evaluate LLaVA-Next-Qwen2 (LongVA without UniRes nor extended LM backbone) on our V-NIAH benchmark. The comparison between LLaVA-Next-Qwen2 and LongVA (AnyRes) in Figure 4 demonstrates the effect of extending the language model’s context length.
>
> The V-NIAH heatmap result is here:
> https://postimg.cc/v4dSXYYd
>
> > [W3]
>
> Thanks for catching those typos! We will update those in the camera-ready version.

---

> > ### Comment · Reviewer_yaCh · 2024-11-26
> >
> > Having reviewed the authors' responses to all reviewers, I find that my concerns have been thoroughly addressed in their detailed replies. I maintain my original rating.

---

### Official Review · Reviewer_92h6 · 2024-11-04

**Soundness:** 2
**Presentation:** 2
**Contribution:** 2
**Rating:** 5
**Confidence:** 5

**Summary:**

This paper proposes a novel way to implement long-context Video MLLM, which only requires long context training of plain text and image-text alignment training.
- The trained long-context LLM is used for image-text training. Through the designed UniRes visual information encoding method, the long-context LLM can be extended to the long-context video MLLM without video-text training.
- A Visual Needle-In-A-Haystavk (V-NIAH) benchmark is proposed to test the capabilities of long-context video MLLM.
- By inputting more frames for reasoning, the proposed LongVA achieves SOTA performance on Video-MME and MLVU.

**Strengths:**

(1) Explored a method to implement long-context video MLLM from the perspective of long-context text training. The idea is very interesting.

(2) The evaluation of long-context text retrieval (NIAH) and video retrieval (V-NIAH) is very complete and well presented.

(3) The experimental results are rich. LongVA was evaluated on different multi-modal benchmarks and achieved SOTA performance on Video-MME and MLVU.

**Weaknesses:**

(1) The results of LongVA on some benchmarks are not ideal, such as ActivityNetQA and VideoChatGPT, which is reflected in the fact that more frames do not show better video understanding ability. This makes people suspect that LongVA only has the ability to retrieve long-context video information, but lacks the ability to understand long-context videos.

(2) The paper does not give detailed training details, such as the training details of the long-context LLM of pure text, the training details of image-text, and the datasets used in these two training processes.

(3) Some experimental results in the article do not show that a long-context MLLM with good video understanding ability has been achieved. And the proposed UniRes has not been proven to be better than the previous AnyRes.

**Questions:**

(1) Can training only be performed on a single image? How is the long text obtained in image-text training? Because the trained LLM backbone has a context length of 224K. The training details are not provided.

(2) According to the display of Figure 2, the author treats each frame of the video as a grid of the image during testing, so 576 tokens are formed for inference. In order to input more frames during inference, my understanding is that during image-text training, the image here needs to have a very high resolution? In order to segment more grids, so as to adapt to more frames during inference. How are high-resolution images obtained? For the collection process of the training set, this seems to be a method of exchanging spatial information annotation for temporal information annotation.

(3) Unless the details of the training are given, including the amount of data, image resolution, and text length, it is difficult to understand why simple image-text training can be transferred to the reasoning of videos with very long frames?

(4) Why can image training achieve long-context video MLLM? The author did not discuss and analyze the principle.

(5) The experimental results in Table 4 show that inputting more frames into LongVA does not continuously enhance the performance of Video-MME? Especially in long duration videos, why is this? And as the number of frames continues to increase, it gets worse in short, medium and long. Although other experiments show that LongVA performs well in VNIAH, demonstrating the ability of information retrieval, it does not show the advantage of this model in long video understanding.

(6) In the experimental results of Table 7, in 6 benchmarks, UniRes is better than AnyRes in 4 tasks, which does not prove that UniRes is a better visual encoding method. Why use UniRes instead of AnyRes?

---

> ### Author Response · Authors · 2024-11-21
>
> We appreciate reviewer 92h6’s valuable suggestions. We have addressed them in detail in the following response.
>
> > [W1]  LongVA’s performance on ActivityNetQA and VideoChatGPT
>
> We offer the following explanations for these observations:
>
> 1. Benchmark Suitability:
> ActivityNetQA and VideoChatGPT primarily consist of relatively short videos (see Table 3). For evaluating long video understanding capabilities, we recommend focusing on benchmarks like VideoMME (Table 4) and MLVU (Table 9), which are better aligned with the objectives of LongVA.
>
> 2. Dataset Bias:
> LLaVA-NeXT-Video models are trained on video datasets generated by ChatGPT, and the evaluation for ActivityNetQA and VideoChatGPT is also conducted using ChatGPT as a judge. This introduces potential bias, favoring models trained on ChatGPT-generated data. Despite this, we include these datasets in good faith to demonstrate that LongVA performs relatively well on them, even without any specific video training.
>
> 3. Performance Improvements with Light Video Training:
>
> After incorporating some video-specific training (which remains significantly lighter than that of LLaVA-NeXT-Video), LongVA-DPO achieves notable performance improvements on these benchmarks. This partially validates our hypothesis stated in Section 2.
>
> > [W2] Training details
>
> Section 3.1 of our paper provides substantial information about the training process for the long-context LLM on pure text.  For information on image-text alignment, please refer to our response to Reviewer 9tBS's Weakness 2, where we detail the pretraining and fine-tuning setups, including datasets and hyperparameters.
>
> We acknowledge the need for further clarity and will include more comprehensive training details in the appendix of the camera-ready version. Additionally, our code and datasets will be open-sourced, ensuring full reproducibility of our results.
>
> > [W3] Video understanding ability and UniRes v.s. AnyRes
>
> We would like to clarify that the primary goal of LongVA is not to achieve state-of-the-art performance but to demonstrate the effectiveness of the long context transfer phenomenon. To this end, we intentionally exclude video training data to highlight that LongVA’s video understanding capabilities are transferred from two sources: (1) its long-text understanding capabilities and (2) its image understanding capabilities. This allows LongVA’s video capabilities to be evaluated in a zero-shot setting.
>
> While excluding video training data provides a clear demonstration of long context transfer, it places LongVA at a disadvantage compared to models specifically trained on video data. This is especially on datasets where response format plays a significant role or when evaluations are judged by ChatGPT. Despite these challenges, LongVA achieves the strongest performance among 7B models on both VideoMME (Table 4) and MLVU (Table 9). We acknowledge that LongVA performs relatively less optimally on certain image datasets and short video datasets, and we believe that incorporating additional image and video data during training could improve its performance. However, this is beyond the primary scope of this paper.
>
> For AnyRes v.s. UniRes, please refer to our response for Q6.
>
> > [Q1] Can training only be performed on a single image? How is the long text obtained in image-text training? Because the trained LLM backbone has a context length of 224K. The training details are not provided.
>
> We can perform training on multi-image and video datasets. However, as the response in W3,  we explicitly exclude the following during image-text alignment to clearly demonstrate the long-context transfer phenomenon:
>
> 1. Multi-image data
> 2. Video data
> 3. Long-text data
>
> This design ensures that LongVA's ability to process long video contexts is entirely transferred from its text-training phase, while image-text alignment focuses solely on adapting the model to process visual signals without introducing long-context information.

---

> ### Author Response · Authors · 2024-11-21
>
> > [Q2] Image grid, high-resolution images, spatial to temporal informatio.
>
> UniRes is designed to unify the representation of images and videos, enabling the transfer of knowledge from spatial to temporal domains. This is a key strength of UniRes, as it allows LongVA to effectively handle video understanding tasks while being trained on images alone. By leveraging this unified representation, LongVA can bridge the gap between image and video modalities without requiring extensive video-specific training.
>
> We believe that the primary factor behind LongVA’s visual length extrapolation capability lies in its training on long-text data, rather than the resolution of the images used. For our training, we utilized the exact same dataset as LLaVA 1.6, which predominantly contains low-resolution images, with the exception of some OCR data. Importantly, no additional high-resolution image data was collected for this work.
>
> To further illustrate this point, the average pixel count of images in the LLaVA-NeXT SFT dataset is 336,759, which corresponds to an image resolution of approximately 580 × 580 pixels. This demonstrates that the UniRes strategy was fully applied to only a small portion of the training data, emphasizing that the success of LongVA is not reliant on high-resolution images but rather on the effectiveness of its unified representation and long-text training.
>
> > [Q3 & Q4] Why image-text training helps long video understanding.
>
> Image training is mainly for aligning the vision and language modalities, and to develop visual understanding capabilities for longva. Image-text training alone is not sufficient for understanding long videos. The key is extending the backbone language model's context length using long text data. This is a separate training phase from image-text alignment. For example, models like LLaVA-Next-Video-32K, which only perform image-text alignment, perform poorly on V-NIAH, as shown in Figure 4
>
> Please refer to our response to Reviewer 9tBS’s Weakness 1 for an intuitive explanation of why long-text training helps long-video understanding.  In brief, this phenomenon arises from the effective modality alignment and the long-context capabilities transferred from text training.
>
>
> > [Q5] No advantage in long video understanding.
>
> LongVA clearly shows its strong performance in long video understanding.  LongVA is able to increase its performance up to 128 frames, after which the model successfully maintains its performance even when extended to 384 frames. We are the first open-source model (outside of closed-source models from companies like Google and OpenAI) to demonstrate that performance can improve as the number of input frames increases and maintain its performance at 384 frames. This is also verified by a concurrent work (Wu, et.al), which shows that existing open-source models fail to understand videos longer than 64 frames. LongVA also achieves the best performance on VideoMME (Table 4) and MLVU ((Table 9) among 7B models.
>
> H. Wu, D. Li, B. Chen, and J. Li, “Longvideobench: A benchmark for long-context interleaved video-language understanding,” arXiv preprint arXiv:2407.15754, 2024.
>
>  We run more experiments on VideoMME of other open-source LMMs:
>
> | Model      | Frames | Short  | Medium | Long  | Total  |
> |------------|--------|--------|--------|-------|--------|
> | LongVA     | 128    | 61.1   | 50.4   | 46.2  | 52.6   |
> | LongVA     | 64     | 61.4   | 50.9   | 45.0  | 52.4   |
> | LongVA     | 32     | 61.1   | 45.4   | 45.4  | 51.8   |
> | IDEFICS2   | 64     | 16.9   | 15.4   | 12.6  | 15.0   |
> | IDEFICS2   | 32     | 55.9   | 49.6   | 42.7  | 49.37  |
>
> **Table: VideoMME Results for LongVA (AnyRes vs UniRes)**

---

> > ### Author Response · Authors · 2024-11-21
> >
> > > [Q5]  Performance improvements on VideoMME end at 128 frames.
> >
> > While long-text training helps transfer long-context capabilities to visual understanding, this transfer does have its limits. For the Video-MME dataset, we observe that performance starts to plateau at around 384 frames. It is neither feasible nor realistic to expect that simply increasing the number of frames will lead to indefinite performance improvement. If this were true, one could theoretically achieve perfect accuracy by continuously increasing the input frames, which is not practical given the inherent limitations of model capacity.
> >
> > Several factors contribute to this phenomenon:
> >
> > 1. **Base Model Capability:** The model's performance is ultimately bounded by the inherent capabilities of its architecture and pretraining.
> >
> > 2. **Relevance of Input Frames:** Many questions in Video-MME pertain to specific durations of a video. Increasing the number of input frames may introduce more irrelevant or redundant information, which the model needs to filter, potentially limiting gains in performance.
> >
> > 3. **Analogy to In-Context Learning:** This behavior is akin to in-context learning in language models. Adding more few-shot examples beyond a saturation point does not lead to further performance improvements and may even confuse the model.
> >
> > Therefore, we believe this observation is intuitive and aligns with our understanding of how contextual capacity transfers from text to video.
> >
> > > [Q6]  AnyRes v.s. UniRes
> >
> >  In Table 7, we honestly report LongVA’s image performance and acknowledge that it performs suboptimally on some datasets. We believe this is due to two main reasons:
> >
> > 1. AnyRes includes an additional base image feature, which may give it an advantage on certain image benchmarks.
> > 2. AnyRes generates more visual tokens for a single image, while UniRes applies 2×2 pooling by default, reducing the number of tokens.
> >
> > However, the primary goal of LongVA is not to optimize image performance but to explore long context transfer and improve video capabilities. UniRes is specifically designed to unify the representation of images and videos. This allows LongVA’s image understanding capabilities to transfer to video, even without video training.
> >
> > As shown in Figure 4, UniRes provides much better length extrapolation capabilities compared to AnyRes. This is why we chose UniRes for LongVA. Furthermore, on the high-resolution InfoVQA benchmark, which requires detailed image understanding with longer input lengths, UniRes significantly outperforms AnyRes. This result further confirms UniRes’s strong long-context extrapolation ability.
> >
> > We further run evaluations on VideoMME of AnyRes v.s. UniRes and report them below:
> >
> > | Model           | Frames | Short  | Medium | Long  | Total   |
> > |------------------|--------|--------|--------|-------|---------|
> > | LongVA (AnyRes)  | 32     | 58.7   | 46.9   | 44.8  | 50.111  |
> > | LongVA (UniRes)  | 32     | 61.1   | 48.8   | 45.4  | 51.8    |
> >
> > **Table: VideoMME Results for LongVA (AnyRes vs UniRes)**

---

> ### Author Response · Authors · 2024-11-27
>
> Dear reviewer 92h6,
>
> Thank you again for finding long context transfer interesting. We are also encouraged by your comments on V-NIAH and our rich experimental results.
>
> We truly value your insights and hope our responses have addressed your concerns. If you have additional feedback or suggestions, we would greatly appreciate hearing them to ensure we have fully resolved the points.

---

> > ### Comment · Reviewer_92h6 · 2024-11-27
> >
> > Thanks to the authors' detailed reply.
> > The author's reply solved some of my questions. But I still have some concerns.
> > - Why is the table in the author's reply to Q5 different from Table. 4 in the paper?
> > - Why does LongVA, after long-context text training and modality alignment, not show long-context ability, i.e., inputting too many frames to obtain more benefits?
> > - The paper's main experiment should be tested using the long video understanding benchmark. Of course, I saw the MLVU benchmark in Table 9. More results should be included.
> > - The experiments with different frames should be conducted on the long video understanding benchmark. For example, VideoMME, MLVU or others. But on the long of VideoMME, no significant benefits from more frames were seen.
> >
> > In general, the author's work aims to verify that the long-context ability of text can be transferred to long-context video understanding without video training. However, I think the experiments and analysis on long-context video understanding are not enough. I choose to keep my score.

---

### Official Review · Reviewer_9tBS · 2024-11-04

**Soundness:** 3
**Presentation:** 2
**Contribution:** 3
**Rating:** 6
**Confidence:** 3

**Summary:**

Current LLMs struggle to understand very long videos because of the context length limitation. This paper addresses this issue by proposing a new method that extends the input context length of LLMs. The method includes three key ingredients: 1) continuous pretraining the LLM with a large context window on text-only dataset, 2) aligning the long-context LLM with image data, 3). tokenizing images (or videos) with the so called UniRes that divides images into grids and flatten the visual tokens in each grid. To evaluate the effectiveness of the method on long video understanding, they introduce a new benchmark called V-NIAH where the task is to retrieve one frame that is randomly inserted into a long video. They also evaluate the method on other video question answering benchmarks. On all the benchmarks, their model "LongVA" demonstrates state-of-the-art performance.

**Strengths:**

1. The proposed method achieves the state of the art on multiple video understand benchmarks.

2. The proposed method is simple and effective. The continuous pretraining on large context window only needs text data, while the multimodal alignment stage only requires image-text data. The uniRes encoding is straightforward extension of AnyRes to videos. Nevertheless, the model seems to generalize surprisingly well to videos.

3. The Needle-in-a-Haystack test seems to be novel in the context of long video understanding. This new task could be an interesting testbed for long video understanding in the future.

**Weaknesses:**

1. Despite the impressive empirical results, there are not much explanations. In particular, I am interested in understanding why the long-context pretraining on text-only data could transfer to long video understanding.

2. It is not clear on how the image-text alignment is done e.g., training data and losses. Most of the content in the Sec. 3.2 is about the UniRes which is relatively straightforward.

3. The number of grids during training is limited to 49 (Line 205), which is effectively equivalent to 49 frames. Why does the model could generalize to handle more than 2K frames give that it is trained with only 49 frame inputs?

4. The results in Table 4 show that LongVA outperforms the methods that are much larger and trained on videos e.g., LLaVA-NeXT-Video-34B. Meanwhile, VILA-1.5 achieves the best results with only 8 frames. Those results seem to imply that the input frames and video-specific training are not that important for video understanding. I found this phenomenon counter intuitive. But this might be caused by the different LLM backbones. One interesting experiment I would suggest is to ablate context length and video specific training with the same LLM.

5. Temporal understanding is arguably an important aspect in video understanding. Being able to process a large number of video frames does not indicate a strong capability in temporal understanding. Most of the experiments shown in the paper can be solved by "selecting" one frame or very few frames. The model might not be able to understand fine-grained motions/actions, for example Something-Something V2 dataset.

**Questions:**

What would be the performance of VILA-1.5 on the V-NIAH benchmark?

---

> ### Author Response · Authors · 2024-11-21
>
> We thank reviewer 9tBS for the insightful comments!  Below, we address the reviewer's concerns in turn.
>
> > [W1] transfer from text-only data to long video understanding.
>
> As stated in line 195 of our paper, we propose the hypothesis: “if the modalities of vision and language are effectively aligned, the capability to handle long contexts could transfer from text to vision.” LongVA builds on this idea by utilizing CLIP, a well-established image model that generates features inherently aligned with text representations. Through further alignment between the LLM and CLIP modalities, we ensure that these representations integrate into the LLM's input space. LongVA is designed as a preliminary effort to test and validate this hypothesis, providing a foundation for further exploration in this direction.
>
> > [W2] Image-text alignment details.
>
> As stated in line 215, we closely follow the training methodology proposed by LLaVA-1.6. Specifically, our setup is as follows:
>
> Pretraining (Feature Alignment):
>
> - Objective: Train only the projection layer connecting CLIP and the LLM while keeping the rest of the model frozen.
> - Loss: Language modeling loss conditioned on the image.
> - Dataset: LAION-CC-SBU 558K with BLIP-generated captions.
> - Hyperparameters:
>   - Learning rate: 1e-3
>   - Batch size: 64
>   - Optimizer: AdamW
>   - Epoch: 1
>
> Visual Instruction Fine-Tuning:
>
> - Objective: Train the projection layer, language model, and ViT model jointly.
> - Loss: Language modeling loss conditioned on the image.
> - Dataset: LLaVA-1.6 (760K samples, image-text pair only).
> - Hyperparameters:
>   - Projection layer and LLM learning rate: 1e-5
>   - ViT learning rate: 2e-6
>   - Batch size: 32
>   - Epoch: 1
>
> We will incorporate this information into the camera-ready version of the paper and provide a detailed training setup in the appendix. Furthermore, our code will be open-sourced to facilitate reproduction.
>
> > [W3] Length generalization of input frames.
>
> Please refer to our response to [W1], which addresses the hypothesis of modality alignment enabling the transfer of long-context capabilities. Specifically, the effective alignment between CLIP and the LLM allows representations to extend beyond the limitations of input frames during training, leveraging the LLM's inherent capability to process longer contexts.
>
> > [W4] Those results seem to imply that the input frames and video-specific training are not that important for video understanding.
>
> We fully agree with the reviewer’s assessment on this matter. We believe VILA-1.5’s high performance could be due to: (1) Some questions in Video-MME may require understanding only a few frames to arrive at the correct answer. (2) There may be biases in the questions and answer choices, allowing a strong language model backbone, such as the 30B+ model used in VILA-1.5, to correctly guess answers without relying on video input.
> These limitations were part of the motivation behind our proposal of V-NIAH. Unlike benchmarks where sparse sampling might suffice, V-NIAH is designed to eliminate this shortcut. Since the needle’s location is not given as prior knowledge, models relying on sparse sampling are likely to miss the needle frame entirely. In V-NIAH, the model must process the entire video (spanning thousands of frames) to locate the relevant information. Additionally, the questions in V-NIAH are intentionally counterintuitive, ensuring the language model cannot rely on textual priors alone and must utilize the video input effectively.
>
> > [W4] Ablation with the same LM backbone of different context lengths and video training.
>
>
> To ablate the impact of the backbone language model’s context length on video understanding, we further run LLaVA-Next-Qwen2 on V-NIAH: https://postimg.cc/v4dSXYYd. Please refer to Table 5 for the relation between LLaVA-Next-Qwen2, LongVA, and LongVA(AnyRes). All three models are trained with exactly the same data and hyperparameters during image-text alignment.
> Our newly added results together with Figure 4 show the text context length of the language backbone (LLaVA-Next-Qwen2 v.s. LongVA(AnyRes)) significantly impacts the visual context length on V-NIAH
>
> Due to limited computational resources, we were unable to include ablation with heavy video training.
> That said, we fully acknowledge that incorporating video training would likely improve the benchmark scores. The decision to exclude video data during training was not due to any disregard for its value, but rather to highlight the long context transfer phenomenon. By omitting video-specific training, we aimed to ensure that LongVA’s performance on video understanding tasks could be classified as zero-shot, offering a clearer validation of the proposed hypothesis.

---

> > ### Author Response · Authors · 2024-11-21
> >
> > > [W5] fine-grained action understanding v.s. visual retrieval capabilities.
> >
> > Thanks for pointing out the distinction between fine-grained action understanding v.s. visual retrieval capabilities. We agree that LongVA and V-NIAH are primarily designed to address the latter. However, we would like to emphasize that locating and retrieving critical information in long videos is far from trivial.
> >
> > As shown in Figure 4 as well as our added results in general response, leading LMMs (LLaVA-Next-32K or LLaVA-Next-Qwen2) struggles to retrieve key information when processing very long video inputs. A concurrent study (Wu et al.) also demonstrates that other open-source video LMMs fail to process more than 64 frames effectively. In contrast, LongVA is the first open-source model capable of handling up to 384 frames on Video-MME and excelling in V-NIAH with inputs spanning 2000 frames.
> >
> > These results highlight LongVA’s significant progress in addressing the challenges of long-context video retrieval. We will explore LonVA’s fine-grained action-understanding capabilities in the future.
> >
> > H. Wu, D. Li, B. Chen, and J. Li, “Longvideobench: A benchmark for long-context interleaved video-language understanding,” arXiv preprint arXiv:2407.15754, 2024.
> >
> > > [Q1] VILA-1.5 on the V-NIAH benchmark.
> >
> > VILA-1.5 is a 34B parameters model, which is significantly larger than LongVA-7B, and we currently lack the computational resources to evaluate it on the V-NIAH benchmark. The V-NIAH benchmark necessitates processing thousands of input frames, which requires substantial GPU memory even beyond 8 × A100 GPUs.

---

### Author Response · Authors · 2024-11-21
**General Response**

We sincerely thank all reviewers for their comments and are encouraged by the shared positive feedback  in the following areas:

1. Introduction of a novel approach that extends the context length of language models for long video understanding, reframing the problem from traditional token reduction.
2. Discovery and analysis of the long context transfer phenomenon, highlighting the transferability of extended context capabilities from language to vision.
3. Development of the V-NIAH benchmark as a valuable tool for assessing long visual context handling in LMMs.
4. Comprehensive experimental evaluations demonstrating state-of-the-art performance on Video-MME and MLVU benchmarks.


We mainly add the following experimental results:

For reviewers’ common concern over AnyRes v.s. UniRes, we further evaluate LongVA(AnyRes) and report them below:

| Model           | Frames | Short  | Medium | Long  | Total   |
|------------------|--------|--------|--------|-------|---------|
| LongVA (AnyRes)  | 32     | 58.7   | 46.9   | 44.8  | 50.111  |
| LongVA (UniRes)  | 32     | 61.1   | 48.8   | 45.4  | 51.8    |

The results demonstrate that UniRes enables better video understanding capabilities in a setup where there is no video during training.

To ablate the impact of the backbone language model’s context length on video understanding, we further run LLaVA-Next-Qwen2 on V-NIAH.
The LLaVA-Next-Qwen2's heatmap result is here: https://postimg.cc/v4dSXYYd.
Please refer to Table 5 for the relation between  LLaVA-Next-Qwen2, LongVA, and LongVA(AnyRes). LongVA, and LongVA(AnyRes)'s results are in Paper Figture 4. All three models are trained with exactly the same data and hyperparameters during image-text alignment.

The results show that increasing the text context length alone has the biggest improvement of visual context length on V-NIAH. Changing AnyRes to UniRes further increases the visual length extrapolation capabilities.

We will address other concerns and questions individually. We will update our manuscript based on the reviewers’ comments.

---

### Meta-Review · Area_Chair_MxJA · 2024-12-21

**Metareview:**

This paper proposes an alternate approach to long-video understanding with VLMs: The authors train the language model with longer textual context lengths, and show how this transfers to the visual modality: ie a language model trained with larger context lengths can perform better on video understanding tasks, even without training on long videos. In addition, the authors also proposed a Visual Needle-in-a-Haystack (V-NIAH) benchmark, and an extension of the Qwen2 VLM that achieves better results on the Video-MME and MLVU datasets.

Reviewers opinions were quite divergent on the paper: Some reviewers appreciated the overall approach which suggests that we don't need video-specific training data, but can simply train with long-context text tasks to perform well for long-context video tasks. However, other reviewers were concerned about the lack of technical novelty, and the fact that although the authors show impressive (and perhaps counterintutive) results on Video-MME and MLVU, there is not much analysis of why the proposed "long context transfer" works. For example, what the failure cases are, in what domains of problems can we expect textual long-context to transfer, and for what domains we actually need video-specifc training data? Although the authors compare to the state-of-the-art in Tables 4 through 7, there is not much analysis of why the proposed method works.

Furthermore, although the authors perform Needle-in-a-haystack experiments, there are also a number of recent works in this area which were not discussed at all [A, B, C, D]. In addition to Video-MME, there are also more recent, real-world video understanding datasets which could also provide more understanding about the "long context transfer" phenomenon. Examples include LVBench (longer, real-world videos) and Egoschema (the videos are not as long, but in contrast to needle-in-a-haystack tasks, require looking at the whole video).

Therefore, after some deliberation, the AC and Senior AC believe that this paper could do with further iteration and encourage the authors to revise the paper in light of reviewers' comments and resubmit to another venue.


[A] Zhao et al. Needle in a video haystack: A scalable synthetic framework for benchmarking video mllms

[B] T Yuan et al. Lv-eval: A balanced long-context benchmark with 5 length levels up to 256k

[C] T Wu et al. Visual haystacks: Answering harder questions about sets of images.

[D] Gemini Team. Gemini 1.5: Unlocking multimodal understanding across millions of tokens of context.

**Additional Comments On Reviewer Discussion:**

Please see above. Reviewers opinions were quite divergent on the paper: Some reviewers appreciated the overall approach which suggests that we don't need video-specific training data, but can simply train with long-context text tasks to perform well for long-context video tasks. However, other reviewers were concerned about the lack of technical novelty, and the fact that although the authors show impressive (and perhaps counterintutive) results on Video-MME and MLVU, there is not much analysis of why the proposed "long context transfer" works. For example, what the failure cases are, in what domains of problems can we expect textual long-context to transfer, and for what domains we actually need video-specifc training data? Although the authors compare to the state-of-the-art in Tables 4 through 7, there is not much analysis of why the proposed method works.

---

### Decision · Program_Chairs · 2025-01-22

Reject